# Analysis of Sexual Disorders in Men with Infrarenal Abdominal Aortic Aneurysm Treated by Stent-Graft or Prosthesis Implantation—A Pilot Study

**DOI:** 10.3390/medicina56040191

**Published:** 2020-04-21

**Authors:** Mariola Sznapka, Anna Brzęk, Damian Ziaja, Michał Tkocz, Krzysztof Pawlicki, Krzysztof Ziaja, Violetta Skrzypulec-Plinta, Jerzy Chudek, Wacław Kuczmik

**Affiliations:** 1Department of General and Vascular Surgery, Angiology and Phlebology Faculty of Katowice, Medical University of Silesia in Katowice, 40-055 Katowice, Poland; m.sznapka@op.pl (M.S.); wkuczmik@sum.edu.pl (W.K.); 2Department of General Vascular Surgery, Faculty of Medicine in Katowice, Medical University of Silesia, 40-659 Katowice, Poland; 3Department of Physiotherapy, Chair of Physiotherapy, School of Health Sciences in Katowice Medical University of Silesia in Katowice, 40-055 Katowice, Poland; dziaja@sum.edu.pl; 4Department of Oncologic and Vascular Surgery, Oncological Centre in Katowice, 40-074 Katowice, Poland; krzysztof.ziaja@op.pl; 5Urologic Department Governor’s Hospital St. Barbara in Sosnowiec, 41-200 Sosnowiec, Poland; tkocz40@interia.pl; 6Department of Biophysics Medical University of Silesia in Katowice, 40-055 Katowice, Poland; kpawlicki@sum.edu.pl; 7Chair of Woman’s Health in Katowice, School of Health Sciences in Katowice, Medical University of Silesia Katowice, 40-055 Katowice, Poland; vskrzypulec@sum.edu.pl; 8Internal and Oncological Department, Medical University of Silesia in Katowice, 40-055 Katowice, Poland; jchudek@sum.edu.pl

**Keywords:** IIEF-5 questionnaire, abdominal aortic aneurysm, sexual readiness, stent-graft implantation, aneurysm surgery

## Abstract

*Background and objectives*: Patients with obstruction or stenosis of the aorta and iliac arteries or with aortic aneurysm, often co-existing with iliac artery aneurysms, suffer from sexual disorders because of insufficient perfusion to the pelvic organs and penis. This is often the cause of visits to a medical doctor’s office with reports of a difficult life situation and a problem with the satisfactory completion of sexual intercourse. A low percentage of vascular surgeons or angiologists are prepared to talk about issues related to the hereditary sphere with a patient who qualifies for the treatment of Leriche syndrome or abdominal aortic aneurysm. The aim of this study was to analyze sexual disorders in men with infrarenal abdominal aortic aneurysm treated by stent-graft or prosthesis implantation. *Material and methods*, Outcomes: 38 patients who completed the IIEF-5 (International Index of Erectile for Men) questionnaire are presented. Initially, 146 qualified for the study after meeting the study inclusion criteria for surgery (Group 1) or for endovascular treatment of abdominal aortic aneurysm (Group 2). *Results*: In the study, no negative impact of smoking was found; however, over 95% of respondents had been smoking for many years in both groups. Patients who qualified for vascular prosthesis implantation were subject to a more advanced atherosclerotic process involving the aorta and iliac arteries. Patients who qualified for stent-graft implantation were twice as often treated for coronary vessel stenosis. In Group 1, the percentage differences, as shown by questions 1 and 5, were statistically significant (58, i.e., 25%, and 40, i.e., 29%). *Conclusions*: Education should target medical personnel in terms of conversations with patients, as well as men who are directly affected by this problem, although their partners and families should not be neglected in these activities. The ability to communicate properly allows for an open dialogue on issues that the patient finds difficult, particularly in the field of sexology.

## 1. Introduction

Patients with obstruction or stenosis of the aorta and iliac arteries or with aortic aneurysm, often co-existing with iliac artery aneurysms, suffer from sexual disorders because of insufficient perfusion to the pelvic organs and penis. This is often the cause of visits to a medical doctor’s office with reports of a difficult life situation and a problem with satisfactory completion of sexual intercourse [1,2,3,4,5,6,7].

This very delicate problem often affects patients aged 50–60, that is, patients with good functional physical fitness who are usually professionally active. In Leriche syndrome [8], although the physiological flow through the internal iliac arteries is closed (in some patients, they fill retroactively), patients with aneurysm are usually not impaired, and the symptoms of sexual dysfunction are similar.

Responding to the question about the self-assessment of sexual function, particularly the completion of a satisfactory vaginal relationship, is perceived as embarrassing and commonly unacceptable to the patient. Before the procedure, a question asked by a vascular surgeon to a patient who is aware of his illness and the complications related to the treatment, including difficulties with intercourse and ejaculation, sometimes results in silence or a very general answer [9,10,11,12,13].

Few patients with a young sex partner ask the question, "when after the procedure will it be possible to have sexual intercourse?" This question is asked very rarely in the presence of a partner. Yet another issue that is rarely raised with doctors is ejaculation. This is a problem that is very difficult from the point of view of the patient’s well-being and acceptance of the problem.

Another element of any subjective self-assessment by the patient before surgery is anxiety caused by objective information on possible adverse events, which are included in the patient’s informed consent sheet, as well as in consultation with the information available from public sources, most often from the Internet. During the hospitalization, the patient receives a great deal of information from other patients before and after the procedure. His objectivity begins to become more and more dependent on collected information, which increases the sense of a threat—anxiety that can distort data collected in surveys, which can be a weakness of the survey [9,10,11,12,13,14,15,16,17]. 

In the authors’ opinion, a low percentage of vascular surgeons, angiologists and phlebologists are prepared to talk about topics related to the hereditary sphere with a patient qualified for treatment for Leriche syndrome [8] or abdominal aortic aneurysm. In the authors’ opinion, it is an extremely important aspect of the patient’s preparation for the procedure and his informed consent and the need to perform such surgery for health reasons.

There is no information about erectile dysfunction in materials regarding questions in the field of vascular surgery or angiology before and after surgical treatment of the above-mentioned pathology. It is only accompanied by the list of symptoms associated with aortic distal aorta obstruction, which is already doubtful at this stage of the education of young doctors. 

For the above-mentioned reasons, the authors were motivated to undertake a pilot study in this area. The aim of this study was to analyze sexual disorders in men with aortic aneurysm of the abdominal aorta treated by stent-graft or prosthesis implantation. The following steps were taken:

1. Assessments of male sexual readiness;

2. Comparative analysis between the tested groups of patients subjected to classical surgery vs. stent-graft implantation;

3. Comparison of the subjective evaluation of the quality of life and level of pain before and after implantation of a vascular prosthesis or stent-graft.

## 2. Materials and Methods

### 2.1. Studied Population

Two-hundred twenty-six patients were treated in 2014 at the Department of General and Vascular Surgery at the Medical University of Silesia in Katowice. Of these 226, 109 qualified for surgery, and 117 qualified for implantation of the stent-graft. Qualifications for the chosen procedure were made in accordance with the Polish Guidelines for the Treatment of Vessel Diseases [15]. The distribution of the examined patients in individual stages is shown in Figure 1.

Finally, the main group included 126 men who met the criteria, of whom 73 (Group 1) were qualified for surgical treatment, and 53 (Group 2) were qualified for stent-graft implantation. In Group 1, there were 49 patients subjected to implantation of a straight prosthesis (group 1A) and 36 patients subjected to implantation of the Y prosthesis (Group 1 B). The IIEF-5 study comprised Gr. 1 with 73 men and Gr. 2 with 53 men. Study inclusion and exclusion criteria are presented in Table 1.

### 2.2. Methods

The assessment of sexual function in men was carried out using the generally available questionnaire “Sexual Health Inventory for Men—IIEF-5”. The questionnaire was distributed among 126 men: Group 1 comprised 73 patients, and Group 2 comprised 53 patients. The characteristics of the respondents included in the final analysis are presented in the results section. The average age of the respondents was 69.3 in Group 1A (between 54 and 83 years; SD 7.36), 68.7 in Group 1B (52–85 years; SD 7.41), and 68.6 in Group B (from 52 to 85 years; SD 7.38).

The questionnaire contained five questions with a maximum score of 5 points for each question for a maximum total score of 25. According to the authors of the questionnaire, a score of 21 or less suggests erectile dysfunction requiring consultation with a doctor.

A subjective assessment of the quality of life was made using the 10-point VAS scale (*Visual Analog Scale*). Patients in the pre-operative stage and patients on the second postoperative day were separately evaluated to obtain a 10-point logarithmic scale for each of the two groups. Patients requiring respiratory treatment answered the questionnaire on the second day after extubation.

### 2.3. Statistical Analysis

Continuous parameters with a normal distribution are presented as average values with SD, and the significance of differences between the groups was assessed by the *t* test. Continuous parameters with a non-normal distribution are presented as medians, and the significance of differences between the groups was assessed by the Mann—Whitney test. Qualitative parameters are presented as percentages, and the significance of differences between the groups was assessed by the χ^2^ test (in the case of expected frequencies, <5 Yates correction was applied). The coefficient of variation % V(X) was determined. A *p* value of 0.05 or less was considered significant. 

### 2.4. Ethical Statement

The study was approved by the Bioethical Committee of the Medical University of Silesia in Katowice under resolution no. KNW/0022/KG/124/14 (3 June 2014). The study conforms to the Helsinki Declaration. All patients provided written informed consent prior to the study, including enrollment and data collection.

## 3. Results

Only 38 questionnaires completed before the procedure were obtained. The questionnaire was returned by 18 patients from Group 1 (24.6%) and 20 patients from Group 2 (37.7%).

A comparison of the average number of points between Group 1 (14.5 pts.) and Group 2 (15.5 pts.) for *n* = 38 on a scale from 0 to 25 pts. showed no statistically significant differences (*p* > 0.83).

### 3.1. Analysis of the IIEF Survey

The individual points of the IIEF-5 questionnaire were analyzed, and there were differences between the groups. 

For question # 1, “How would you rate your confidence that you could have and keep an erection of penis”, on a scale of 1–5, the average score in Gr. 1 was 2.35 (in the survey: “low”), while the Gr. 2 avg. was 2.85 (in the survey: “moderate”); the confidence interval was 58% in Gr. 1 to 25% in Gr. 2.

For question # 2, “When you did have an erection during sexual stimulation, how often was your erection strong enough to penetrate”, on a scale of 0–5, the mean score in Gr 1 and Gr 2 was 2.6 (in the survey: “several times”, “much less than half the time”), with a confidence interval of 50% vs. 48%.

For question # 3, “During sexual intercourse, how often did you have the ability to keep your penis erect (not revived) after entering the vagina of your partner”, the average in Gr 1 was 2.3 vs. 2.5 in Gr 2 on a scale of 0–5 (in the survey: “several times”, “much less than half the time”), with a confidence interval of 50% vs. 44%.

For question # 4, “During sexual intercourse, how difficult was it for you to maintain a penile erection until the end of the intercourse”, the average in Gr. 1 and 2 was 2.8 on a scale from 0 to 5 (in the survey: from the determination, “it is very difficult”), with a confidence interval of 50% vs. 48%.

For question # 5, "When you tried to have sexual intercourse, how often was it satisfactory for you", the average in Gr. 1 was 2.95 on a scale from 0 to 5 (in the survey, similar scores for “sometimes”, “about half the time”), and in Gr. 2, the average was 3.4 (in the survey, a trend of “many times”, “much more than half the time”), with a confidence interval of 40% vs. 25%. (cf. Table 2)

From the 126 questionnaires sent to male patients with a returnable envelope enclosed (30–60 days after the patient was discharged), we received 6 incomplete questionnaires with inadequate comments. We are almost certain that the decision about returning the questionnaire was a family one, and additionally, in Silesia, decisions about the home are made by women. Additionally, the authors are aware that these patients are those suffering from arteriosclerosis. There were no differences in the average age range or the use of stimulants, particularly cigarette smoking (cf. Table 3 and Table 4). 

In Group 2, a higher percentage of patients with coronary disease after coronary artery bypass grafting (CABG) and percutaneous coronary intervention (PCI) and after myocardial infarction was observed (cf. Table 5).

### 3.2. Assessment of Quality of Life and Pain

An interesting observation is that both groups reported the same level of pain. Before the procedure, the reported pain level was 1.2–2.1 (avg. 1.8; SD 0.9); on the second day after the procedure of implantation of the prosthesis or stent-graft, the reported level of pain was 6.0–7.5 (avg. 6.77; SD 1.5), despite the fact that the patient leaving the hospital after implantation of the stent-graft was much more physically fit and had a smaller surgical wound.

Before the procedure, patients in both groups (1 and 2) assessed their quality of life to be at a very low level, with an average score of 1.04 (SD 1.8; range 0.5–2.3). The reason that they provided was a high risk of rupture of the aneurysm, which would cause their immediate death.

On the second day after the surgery, despite a significantly higher level of pain, they rated their quality of life as very high, with both groups scoring above 6 pts. In Group 1, the average was 7.9 (SD 4.9; range 5.1–10), and in Group 2, the average was 6.4 (SD 3.6; range 4.8–8.4).

## 4. Discussion

Sexual dysfunction is becoming a less embarrassing subject of physician visits; this condition is associated with a reduced quality of life and family problems, which are sometimes the cause for its disintegration [6,7,8,10,11]. However, in the opinion of the authors, it is still insufficiently addressed.

The perception that erectile dysfunction is only an impaired flow through the internal iliac arteries seems, in the opinion of the authors, an oversimplification of this complex problem involving the inflow, perfusion and outflow of blood from the penile organ, i.e., the penis [8,14,15,18]. The authors are rather convinced that despite a thorough explanation to the patient of the questionnaire by the professor, it is not comprehensible to most patients. The authors have not managed to change the nomenclature of the questionnaires (standardization of the research tool), so it may have been incomprehensible to our patients. The authors are aware that sexuality in the 60+ group is poorly understood, but this is not a reason to not examine them. Shirii R et al. reported that after the age of 70 years, the percentage of men with erectile dysfunction (ED) in the general population is estimated to be 75% [19]. Eardley I et al. [20] observed the sexual habits of British men and women over 40 years old, and Rosen RC [21] estimated a less widely appreciated association between erectile dysfunction and other common urological disorders in aging Australian men. Because there were no reports after surgery, the authors considered this problem to be extremely important in terms of widely defined determinants of successful aging. The problems related to men’s sexuality in Poland are rarely discussed, especially in seniors. This is because this group is often very religious. However, many issues, especially in the regions of Silesia, are decided by women, which could be an important factor in the results obtained. Because these questions are outside of patients’ comfort zones, and all of them can be seen by patients as embarrassing, it is essential to ensure full discretion and seriousness when answering them.

Chronic ischemia of the pelvic organs is the cause of a number of changes caused by the intensification of ischemia during physical exercise—gluteal claudication—and the conversion to anaerobic metabolism, endothelial damage and thereby the saturation of organs by diffusion, including nerve fibers of the pelvis directly innervating the penis.

Another problem is the technique for preparing iliac vessels for their bifurcation—closure of the median sacral artery and lumbar arteries—in patients treated with both types of procedures [15,22,23,24,25,26]. 

The authors compared the IEFF-5 questionnaire with the readiness to have sexual intercourse in two groups of patients with aortic subcutaneous aneurysms. However, the study has some limitations: the compared groups were not completely homogeneous in terms of the extent of the atherosclerotic process of the aorta in Group 1, with a significantly higher proportion of patients with coronary atherosclerosis in Group 2 (cf. Table 2).

Patients qualified for stent-graft implantation have to meet anatomical requirements (the infrarenal segment of the aorta should be ≥ 15 mm, the width should not exceed 32 mm, the common iliac artery ≤18 mm), whereas the surgical procedure may be performed on patients who have concomitant common iliac artery aneurysms [1,2,5,12,13,16,27,28,29,30,31].

An additional element that is very difficult to assess is a change caused by damage of the endothelium, a pronounced increase in mediators of inflammation or pro-angiogenetic generation processes in the aneurysm neck in relation to the increased expression of TNF gene mediators and activation of the apoptosis pathway or planned cell death in its distal part, particularly in patients with concurrent aneurysms of the iliac arteries in conjunction with impaired vascular remodeling and angiogenesis, which is decisive in the possibility of nourishing the wall from the outside and not by diffusion in these patients. Whether this is the result of the degree of endothelial damage to the corpora cavernosa requires further investigation [18,21,22,32].

The authors based their considerations on a relatively small group of patients who submitted completed questionnaires, which further suggests that patients still consider this issue to be embarrassing. Greater sexual readiness was found in patients with a more advanced process of atherosclerosis in Group 1, in comparison with Group 2, with a more advanced disease process involving coronary vessels.

In the patients studied, age and stimulants seemed to have little importance. The negative impact of smoking on the presented sexual readiness has not been proven, but it should be emphasized that, in both groups, over 95% of patients were long-term cigarette smokers; thus, the denial or confirmation of the role of chronic nicotinism was impossible [16,24,33,34]. The limitations of the current study include its cross-sectional design and cohort size. Therefore, further research is needed to understand this specific problem. Because of the insufficient number of completed questionnaires and, most importantly, the lack of response to post-surgical observations, the authors do not draw specific conclusions but only indicate a problem. It is the authors’ belief that in the didactic process of vascular surgeons, urologists, angiologists and phlebologists, the emphasis should be on education regarding the sphere of this problem [35]. Therefore, further research is needed to understand the individual and social impact of the problem on individual patients and on the quality of healthcare provision. Future studies should not only be carried out on larger scales but also take into account other factors connected to sexual disorders in men.

## 5. Conclusions

Education should cover both medical personnel in terms of conversations with patients, as well as men who are directly affected by this problem, although their partners and families should not be neglected in these activities. The ability to communicate properly allows for an open dialogue on issues that are difficult for the patient, particularly in the field of sexology.

## Figures and Tables

**Figure 1 medicina-56-00191-f001:**
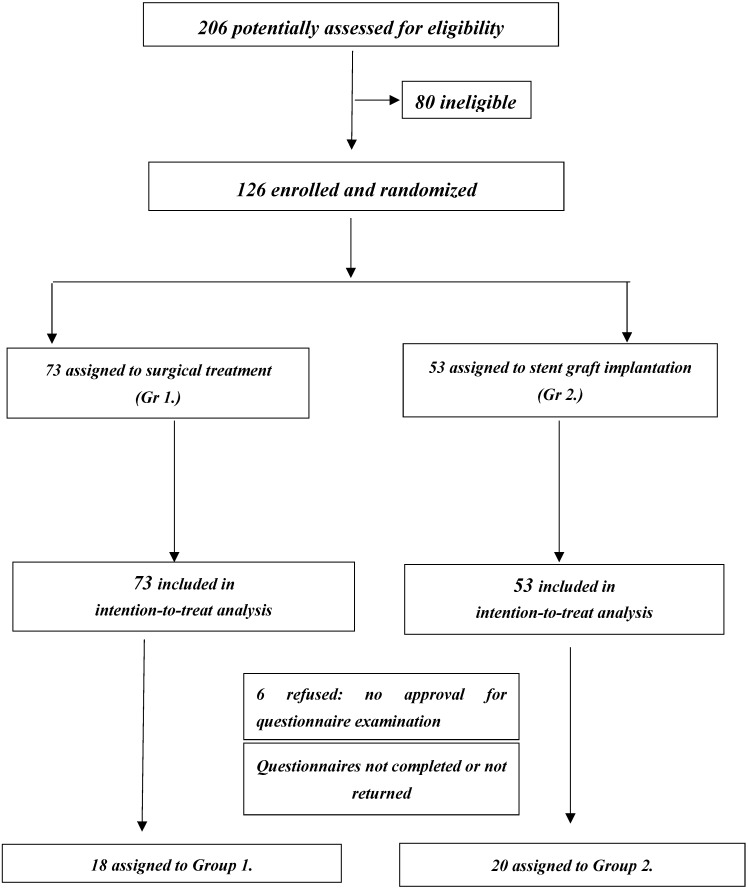
Visual insight into the selection procedure of participants (flow diagram).

**Table 1 medicina-56-00191-t001:** Study inclusion and exclusion criteria.

Inclusion Criteria	Exclusion Criteria
✓ Diameter of AAA >55 mm in M, Expanding AAA >10 mm/year or 5 mm in 6 m. ✓ Pain of the AAA during physical examination ✓ Prosthesis implantation included patients disqualified for stent-graft procedure ✓ Patients with right anatomical criteria were qualified for the study ✓ Life expectancy <10 years	✓ Recommendation for closing or covering one of the iliac arteries because of denervation ✓ Recommendation for aorta bifemoral prosthesis implantation ✓ Stenosis one or both ICA >70% or previous stroke/TIA ✓ Recommendation for pacemaker implantation ✓ Pulmonary obturator diseases ✓ Disseminated neoplasm ✓ Chronic renal insufficiency, creatinine level >1.5 mg% ✓ Thoracic abdominal aneurysm or ruptured aneurysm

Abbreviations, AAA—abdominal aortic aneurysms; ICA—internal carotid artery; TIA—transient ischemic attack.

**Table 2 medicina-56-00191-t002:** Coefficient of variation % V(X) for *n* = 38.

Variable	Group 1; *N* = 18 Mean (SD)	Group 2; *N* = 20 Mean (SD)	V (X) W %
IIEF 5” = 25 p	(*n* = 4) 14.5 (3.81)	(*n* = 3) 14.1 (3.75)	Gr. 1 = 26 Gr. 2 = 26
IIEF ≤ 21	(*n* = 14) 19.42 (4.40)	(*n* = 17) 15.22 (3.88)	Gr. 1 = 22 Gr. 2 = 25
Q 1	2.35 (1.38)	2.85 (0.72)	Gr. 1 = 58 Gr. 2 = 25
Q 2	2.6 (1.3)	2.6 (1.21)	Gr. 1 = 50 Gr. 2 = 48
Q 3	2.3 (1.2)	2.5 (1.12)	Gr. 1 = 50 Gr. 2 = 44
Q 4	2.8 (1.4)	2.8 (1.37)	Gr. 1 = 50 Gr. 2 = 48
Q 5	2.95 (1.2)	3.4 (1.0)	Gr. 1 = 40 Gr.2 = 29

Abbreviations, Q—question.

**Table 3 medicina-56-00191-t003:** Comparison of percentages of participants in both groups of analyzed parameters.

Variable	Group 1; *N* = 73	Group 2; *N* = 53
Smoking, *n* (%) (20–30 daily, 30–40 years)	72/73 (98.63%)	50/51 (94.33%)
Coffee (per day)	2–4	2–4
Alcohol	Occasionally	Occasionally
Claudicatio intermittens < 200 m	27.4 %	30.18 %
Varices veins	42.46 %	64.15 %
Pulmonary obturator diseases	15.06 %	39.62%

Abbreviations, yr—year; m—meter.

**Table 4 medicina-56-00191-t004:** Comparison of percentages of participants in all groups of analyzed comorbidities.

Variable	Group 1A; *N* = 49 * (%)	Group 1B; *N* = 36 ** (%)	Gr 2; *N* = 61 *** (%)
Hypertension, (%)	95.9	94.4	96.7
CABG	12.2	19.4	22.9
MI, n (%)	12.2	5.5	24.5
CAD	18.3	33.3	52.4
Diabetes, (%)	18.3	11.1	16.3

Abbreviations, MI—Myocardial infarction in medical history; CABG—coronary artery bypass grafting; CAD—coronary artery diseases; Group 1A—straight prosthesis; Group 1B—implantation of the Y prosthesis; * the distal anastomosis straight prosthesis is located over aortic bifurcation; ** distal anastomosis of bifurcated prosthesis is located over iliac artery bifurcation, *** fixation of the distal bifurcated stent graft arm is over iliac artery bifurcation.

**Table 5 medicina-56-00191-t005:** Comparison of percentages of participants in all groups of analyzed clinical parameters and comorbidities of subgroups with implantation of a straight prosthesis (Group 1A) and patients with implantation of the Y prosthesis (Group 1B); based on completed IIEF-5 questionnaire.

Variable	Surgery	Stent Graft Implantation	*p*
Patient age, years	68.5 (Range 53–85)	68.7 (Range 52–83)	NS
Cigarettes, %	94.4%	95.0%	NS
Coffee, Range	2–4	2–4	
Alcohol	Occasionally	Occasionally	
Hypertension, %	94.4%	95.0%	NS
Diabetes, %	11.1%	10.0%	NS
CAD, %	16.6%	45.0%	*p* < 0.05
MI, %	11.1%	15.0%	NS
CABG, %	11.1%	20.0%	*p* < 0.05
Claudicatio intermittens <200 m	22.2%	25.0%	NS
Varices and pulmonary embolization	44.4%	45.0%	NS

Abbreviations, MI—Myocardial infarction in medical history; CABG—coronary artery bypass grafting; CAD—coronary artery diseases.

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
