# Peer review of "Analysis of Sexual Disorders in Men with Infrarenal Abdominal Aortic Aneurysm Treated by Stent-Graft or Prosthesis Implantation—A Pilot Study"

_medicina, 2020, doi:10.3390/medicina56040191_

Round 1
Reviewer 1 Report
Major
The manuscript is not well designed and executed and, apart the need about education on sexual dysfunction after aortic aneurysm surgery for both physicians and patients, the aims of the study remain unclear.
Moreover, in my humble opinion, the cohorts are too small to come to any conclusions.
Whilst not wishing to detract from sexual function in the elderly, after the age of 70 ys in the general population the percentage of man with erectile disfunction (ED) is estimated in 75%. (see. Shiri R et al. Prevalence and severity of erectile dysfunction in 50 to 75 year old Finnish men. J Urol 2003; 170: 2342–2344., Eardley I et al. The sexual habits of British men and women over 40 years old. BJU Int 2004; 93: 563–567 and Rosen RC. Reproductive health problems in ageing men. Lancet 2005; 366: 183–185).
So, regardless the concerns about sexual disfunctions after surgical correction of abdominal aortic aneurysm (AAA), ED is rather common in older people. A stratification between different decades is a crucial issue but the samples here presented are too small.
Moreover, the analysis is focused on patients submitted to surgery for AAA that is a rather different pathology from Aorto-iliac steno-occlusion (AKA Leriche’s Syndrome) discussed in the introduction. In the latter case, sexual impairment is a rule due to arteriopathy of the hypogastric artery and its distal branches, while, in the former, it should affect less patients, not considering the above described prevalence among the elderly.
Instead, AAA’s patients could suffer from sexual disorders after surgery. In case of Open Aortic Repair (OAR) through a translaparotomic access, full aortic exposure, cross clamping and insertion of a graft, the rate of sexual dysfunction approximates 50% (and may be as high as 80%). It is mainly related to the dissection along the anterior surface of the aorta and, in particular, over the left iliac artery, with injuries of the neural bundles running from the spinal cord, resulting in diminished or eliminated innervation.
On the contrary, EVAR allow to spare the neurological damage but exposed the patient to the risk of hypogastric arteries coverage in case of distal landing to the external iliac arteries. Obviously, EVAR is a less invasive intervention that is usually suggested for higher risk patients, reduce immediate post-operative complications and could allow a quicker and simpler post operative recovery that can explain the slightly better results of IIEF 5” in the EVAR group as already reported by Prinssen et al, where after EVAR, recovery to preoperative levels is faster than after OAR but at 3 months, sexual dysfunction levels are similar in both groups. (see Prinssen M. et al, Sexual Dysfunction After Conventional and Endovascular AAA Repair: Results of the DREAM Trial. J ENDOVASC THER 2004;11:613–620).
Minor
Please, use Open Aortic Repair (OAR) instead of surgical treatment and EndoVascular Aortic Repair (EVAR) instead of stent graft implantation.
In the Matherial and Methods section the authors should preliminarly cut any female case outside from the analysed population.
In the Results section (3.2) there is a description of 2-days post operative pain that has little to do with the aim of their analysis.
Author Response
Dear Sir or Madam
Reviewer # 1
Medicina Lithuania Editorial Office
Thank you for your help in prolonging the deadline for manuscript correction. I wish to resubmit a cover letter manuscript entitled: “ Analysis of sexual disorders in men with infrarenal abdominal aortic aneurysm treated by stentgraft or prosthesis implantation” in which it has been described in details how we have followed the reviewers suggestions. In manuscript we changed the author’s other.
Below, we have answered the Reviewer’s comments. Additionally, we attached the new version of the manuscript. We have marked in red and highlighting the responses to each point of reviews and all of the revisions made in the text. We hope that in its amended form, the article is now suitable for publication in Medicina Lithuania.
Reviewer (#1),
We would also like to thank the Reviewer for their your comments. We are so glad of your opinion.
- The manuscript is not well designed and executed and, apart the need about education on sexual dysfunction after aortic aneurysm surgery for both physicians and patients, the aims of the study remain unclear.
Moreover, in my humble opinion, the cohorts are too small to come to any conclusions.
Answer: Indeed, in the end, there are quite a few groups, which is additionally marked in the weaknesses of our work. We pointed out the need to analyze other factors influencing sexuality disorders in men (see these line 228-244, p. 8). The authors wrote a sentence “The above premises prompted the authors to undertake pilot studies in this area”(see these line 94, p. 1). And the pilot study has been added to the title of the manuscript. As suggested, the objectives have been clarified (see these line 94-100, p. 1 &2).
- Whilst not wishing to detract from sexual function in the elderly, after the age of 70 ys in the general population the percentage of man with erectile disfunction (ED) is estimated in 75%. (see. Shiri R et al. Prevalence and severity of erectile dysfunction in 50 to 75 year old Finnish men. JUrol 2003; 170: 2342–2344., Eardley I et al. The sexual habits of British men and women over 40 years old. BJU Int 2004; 93: 563–567 and Rosen RC. Reproductive health problems in ageing men. Lancet 2005; 366: 183–185).
Answer: The authors are aware that sexuality in the 60+ group is hardly understandable, but it is not a reason why we should not examine them. And due to the fact that there are no reports after surgery, the authors considered this problem to be extremely important in terms of widely defined determinants of successful aging The Reviewer’s chapter have been completed on page no( see these line 228-244, p. 8). Thank you so much.
- Moreover, the analysis is focused on patients submitted to surgery for AAA that is a rather different pathology from Aorto-iliac steno-occlusion (AKA Leriche’s Syndrome) discussed in the introduction. In the latter case, sexual impairment is a rule due to arteriopathy of the hypogastric artery and its distal branches, while, in the former, it should affect less patients, not considering the above described prevalence among the elderly.
Instead, AAA’s patients could suffer from sexual disorders after surgery. In case of Open Aortic Repair (OAR) through a translaparotomic access, full aortic exposure, cross clamping and insertion of a graft, the rate of sexual dysfunction approximates 50% (and may be as high as 80%). It is mainly related to the dissection along the anterior surface of the aorta and, in particular, over the left iliac artery, with injuries of the neural bundles running from the spinal cord, resulting in diminished or eliminated innervation.
On the contrary, EVAR allow to spare the neurological damage but exposed the patient to the risk of hypogastric arteries coverage in case of distal landing to the external iliac arteries. Obviously, EVAR is a less invasive intervention that is usually suggested for higher risk patients, reduce immediate post-operative complications and could allow a quicker and simpler post operative recovery that can explain the slightly better results of IIEF 5” in the EVAR group as already reported by Prinssen et al, where after EVAR, recovery to preoperative levels is faster than after OAR but at 3 months, sexual dysfunction levels are similar in both groups. (see Prinssen M. et al, Sexual Dysfunction After Conventional and Endovascular AAA Repair: Results of the DREAM Trial. J ENDOVASC THER 2004;11:613–620).
Answer: Thank you for this opinion, but the distal anastomosis straight prosthesis is located over aortic bifurcation and distal anastomosis of bifurcated prosthesis is located over iliac artery bifurcation and fixation of the distal bifurcated stentgraft arm is over iliac artery bifurcation. This information have been added under table No. 4 ( see these line 199-201, p. 7). And exclusion criteria has been completed in point No. 1 (cf. Tab. 1, p. 5)
- Please, use Open Aortic Repair (OAR) instead of surgical treatment and EndoVascular Aortic Repair (EVAR) instead of stent graft implantation.
Answer: The authors used such operating techniques as described in the methodology. Based on the knowledge and experience of operators working on the subject in practice
- In the Matherial and Methods section the authors should preliminarly cut any female case outside from the analysed population.
Answer: Thank you for this suggestion. This is due to the fact that research is part of a large research project in which women also participate. Women were removed from this study (in the text and the tables and fig.).
- In the Results section (3.2) there is a description of 2-days post operative pain that has little to do with the aim of their analysis.
Answer: The authors presented the results of the second day's pain due to the fact that patients from the Intensive Care Unit (ICU) were transferred, especially after stentgraft implantation.
- The manuscript has been proofreading by the professional Native English speaker. We put the English certificate in Manager System
Thank you for your consideration of this manuscript.
Sincerely,
Anna Brzęk, Assoc. Prof.
Reviewer 2 Report
The authors around M. Sznapka attempt the analysis of sexual disorders in men
with infrarenal abdominal aortic aneurysm treated by stentgraft or prosthesis implantation. While the introduction and results seems quite clear, in my opinion attention should be given to the description of the methodology which has been explained but needs some improvements.
The main flaw of the study is represented by the section Materials and methods, in particular the subsection population. For example, it is not clear the role of the females enrolled in the study. The authors stated that 146 candidates met the criteria (126 men and 20 women). However, the work seems oriented and written towards sexual problems in males, or at least this is the main impression when reading. The authors should try to clarify this point. In the section Methods it is written that the questionnaire is given to 126 men.
In the Results section (Analysis of the IIEF survey) the cited questions would allow for a deeper discussion than that given. For this reason, in my opinion, the section Discussion could be improved. In general, the embarass of patients for talking about their sexual problems for one side and the endovascular treatments they underwent from the other side is sometime confusing and diffuclt to interpret and understand. I would recommend the authors to give more details and organize better the description of the results.
The study is interesting and makes a contribution. However, a better description of the Methods and a deep Discussion is required.
Author Response
Dear Sir or Madame
Reviewer #2
Medicina Lithuania Editorial Office
Thank you for your help in prolonging the deadline for manuscript correction. I wish to resubmit a cover letter manuscript entitled: “ Analysis of sexual disorders in men with infrarenal abdominal aortic aneurysm treated by stentgraft or prosthesis implantation” in which it has been described in details how we have followed the reviewers suggestions.
Below, we have answered the Reviewer’s comments. Additionally, we attached the new version of the manuscript. We have marked in red and highlighting the responses to each point of reviews and all of the revisions made in the text. We hope that in its amended form, the article is now suitable for publication in Medicina Lithuania. In manuscript we changed the author’s other. We put English Certificate in Manager System.
Reviewer (#2),
We would also like to thank the Reviewer for their comments, which helped us to rewrite and improve the manuscript.
- The main flaw of the study is represented by the section Materials and methods, in particular the subsection population. For example, it is not clear the role of the females enrolled in the study. The authors stated that 146 candidates met the criteria (126 men and 20 women). However, the work seems oriented and written towards sexual problems in males, or at least this is the main impression when reading. The authors should try to clarify this point.
Answer: Answer: thank you for this suggestion. This is due to the fact that research is part of a large research project in which women also participate. Women were removed from this study (text and tables and fig.).
- In the section Methods it is written that the questionnaire is given to 126 men.
In the Results section (Analysis of the IIEF survey) the cited questions would allow for a deeper discussion than that given. For this reason, in my opinion, the section Discussion could be improved. In general, the embarass of patients for talking about their sexual problems for one side and the endovascular treatments they underwent from the other side is sometime confusing and diffuclt to interpret and understand. I would recommend the authors to give more details and organize better the description of the results.
Answer: The discussion section has been completed. (see these line 228-244, p. 8). Thank you so much.
- The manuscript has been proofreading by the professional Native English speaker.
- The reference has been completed and all have been checked
Please address all correspondence concerning this manuscript to me at: aniabrzek@interia.pl
Thank you for your consideration of this manuscript.
Sincerely,
Anna Brzęk, Assoc. Prof.
Round 2
Reviewer 1 Report
Dear Editor,
I have reviewed the manuscript Medicina-737884 ANALYSIS OF SEXUAL DISORDERS IN MEN WITH INFRARENAL ABDOMINAL AORTIC ANEURYSM TREATED BY STENTGRAFT OR PROSTHESIS IMPLANTATION of Mariola Sznapka, Damian Ziaja, Michał Tkocz, Krzysztof Pawlicki, Anna Brzęk *, Krzysztof Ziaja, Violetta Skrzypulec-Plinta, Jerzy Chudek, Wacław Kuczmik.
The authors have provided a reviewed version replying to my comments.
I note with satisfaction that also the title has been modified adding the definition of “Pilot Study”. In this perspective, any weakness here in observed become now more acceptable.
Moreover, many amendments to the original manuscript have been adopted thus allowing to consider this manuscript worthy of interest for publication.
Anyway I still suggest to use the terms of Open Aortic Repair (OAR) or Opean Aortic Surgery (OAS) instead of "surgical treatment" but above all EndoVascular Aortic Repair (EVAR) instead of stent graft implantation.
Best regards.
Reviewer 2 Report
The authors have responded to the concerns highlighted during the review so that I am pleased to recommend the manuscript for publication.